# Detecting shortcut learning for fair medical AI using shortcut testing

Alexander Brown[1], Nenad Tomasev [2], Jan Freyberg[3], Yuan Liu[4],
Alan Karthikesalingam[3] & Jessica Schrouff [2] ✉

Machine learning (ML) holds great promise for improving healthcare, but it is critical to ensure that its use will not propagate or amplify health disparities. An important step is to characterize the (un)fairness of ML models–their tendency to perform differently across subgroups of the population–and to understand its underlying mechanisms. One potential driver of algorithmic unfairness, shortcut learning, arises when ML models base predictions on improper correlations in the training data. Diagnosing this phenomenon is difficult as sensitive attributes may be causally linked with disease. Using multitask learning, we propose a method to directly test for the presence of shortcut learning in clinical ML systems and demonstrate its application to clinical tasks in radiology and dermatology. Finally, our approach reveals instances when shortcutting is not responsible for unfairness, highlighting the need for a holistic approach to fairness mitigation in medical AI.

Machine learning (ML) promises to be a powerful approach in many healthcare settings, with models being designed for a variety of diagnostic and prognostic tasks. Risk of harm from machine learning models is unfairness, as variation in model behavior for patients with different sensitive attributes (Fig. 1a) has the potential to perpetuate or amplify existing health inequities[1,2]. This has been observed in multiple clinical settings[3–5] and remains a major topic of research. While the definition of what constitutes fairness may vary widely across fields[6], here we focus on the expression of fairness as equal model performance across patient subgroups defined by sensitive attributes[7].

On the other hand, machine learning models may utilize information about sensitive attributes (such as age, sex, or race) to improve model performance[8] in ways that may be justifiable where attributes correlate with disease risk/presentation in the deployment population. For instance, androgenetic alopecia is more prevalent in men and breast cancer more common in women; keloid scarring is more common in skin of color[9], and melanoma is more common in lighter skin tones. In such settings, ignoring or ablating attribute information may decrease clinical performance.

However, the use of information about sensitive attributes can also be harmful–in particular, due to the phenomenon of shortcut learning[10]. This refers to ML models relying on spurious associations in training datasets to learn prediction rules which generalize poorly, particularly to new populations or new settings. Shortcut-based decision rules are also likely to amplify errors in atypical examples, such as male patients with breast cancer or melanoma in dark-skinned individuals. While shortcut learning is typically evaluated by focusing on the performance of a model in different populations or environments[10], shortcuts based upon sensitive attributes have the risk of exacerbating model unfairness and to further health disparities. Our work hence investigates how shortcuts might affect model fairness, in addition to performance.

Concerns about ML models exploiting shortcuts based on sensitive attributes have been amplified by the observation that ML models can predict these attributes from clinical data without the need for attributes to be directly inputted into the model. For example, models can be trained to predict sex or age from medical images[11,12] and may even encode information about sensitive attributes when this was not the objective of ML training[13]. However, the fact that a ML model encodes information about sensitive attributes does not necessarily mean that it uses this information to make clinical predictions[14] or that such use results in shortcut learning.

Previous work on shortcut learning[10] has focussed on sensitive attributes that are likely to encode spurious correlations. In this

[1]UCL Institute of Child Health, London, England. [2]Google DeepMind, London, UK. [3]Google Research, London, UK. [4]Google Research, Palo Alto, CA, USA. Work performed while at Google Research: Alexander Brown. ✉e-mail: schrouff@google.com

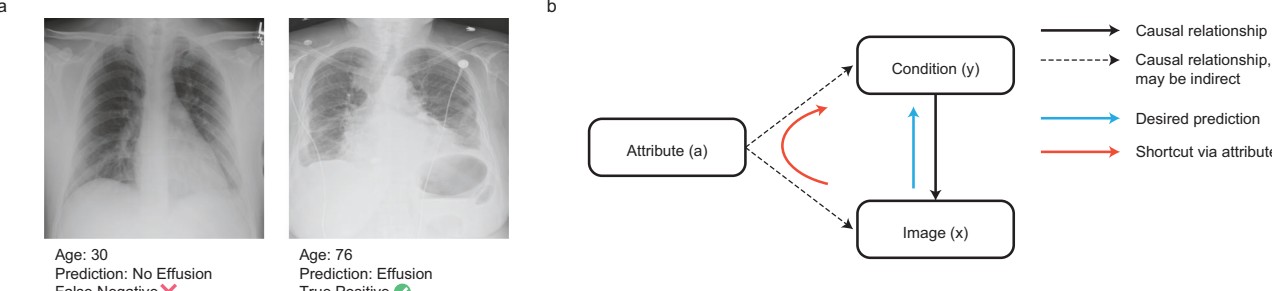

**Fig. 1 | Is a model unfair due to shortcutting? a** Examples of correct and incorrect predictions that may be influenced by age shortcut learning in a chest X-ray application detecting Effusion. In this example, the prediction is incorrect for a patient with atypical age, raising the possibility of shortcut learning. **b** Simplified diagram illustrating how shortcutting may occur. In this schematic, the presence or absence of a particular condition *y* will produce changes in the image *x*; we therefore wish to train a model that can predict *y*, given *x* (blue arrow). However, an attribute *a*, such as age, may alter both the risk of developing a given condition, as well as the image. This need not be a directly direct relationship (dotted arrows). A model may learn to predict the presence of a condition by using the attribute (red arrow). When these correlations are not considerably beneficial for model performance, we consider that this path is (at least partly) a shortcut.

context, any reliance by the model on the sensitive attribute may be considered to be a shortcut. However, this approach does not generalize to cases where a sensitive attribute may be, at least partly, causally related to the outcome. In this work, we posit that the effect of sensitive attributes on the model is the sum of biological, potentially causal effects that might improve model performance, and shortcut learning, which might be harmful. In this context, we redefine shortcut learning to be an effect of the sensitive attribute that does not considerably improve performance (as defined by the user) but affects fairness. By intervening on the degree to which a model can encode a sensitive attribute, we demonstrate a method that assesses whether such encoding indicates the presence of shortcut learning, appropriate use of sensitive attributes, or is artefactual.

The main contribution of this study is an approach that represents a practically applicable framework for studying and mitigating shortcut learning in clinical ML models. This addresses an unmet need among practitioners when trying to develop fair and safe clinical AI. To illustrate our method, we refer to radiology and dermatology applications and focus on age as a sensitive attribute since aging is linked to disease risk across a wide variety of medical conditions, making it harder to reliably establish whether a model is relying on shortcuts. In addition, we follow up on prior work by Gichoya et al.[13] investigating the effect of race encoding to understand how this encoding affects model performance and fairness. In this case, we consider race to be a social construct akin to a spurious correlation[13].

## Results

### Prediction models encode age and are unfair

Using an open-source chest X-ray (CXR) dataset, we trained separate deep learning models for each of the three CXR prediction targets (Atelectasis, Effusion, Abnormal, Supplementary Fig. 1a). The model architecture comprised a feature extractor followed by a clinical prediction head[8]. Using a transfer learning paradigm (i.e., freezing the weights of the feature extractor and training the model to predict age), we then characterized the amount of age information contained in the penultimate layer of each model (see "Methods").

We find that transfer models were able to predict age (Supplementary Fig. 1b; Effusion $11.9 \pm 0.47$ years; Atelectasis $11.3 \pm 0.28$; Abnormal $11.4 \pm 0.44$, age MAE on the held-out test set) significantly better than chance (permutation test, $p < 0.0001$ for all models). We estimated experimental bounds on these results by training the full architecture to predict age (lower error bound) and to predict the mean age across training samples (upper error bound, see "Methods"). Models performed better than the upper error bound of 13.6 years but worse than the lower error bound defined by a direct prediction model of $6.4 \pm 0.23$ years.

We then estimated algorithmic fairness as defined by separation[7], i.e., discrepancies in the model's true positive and false positive rates based on age. All tasks produced models with a bias in performance according to age based on separation (Supplementary Fig. 1c). The observed separation values, in the range of 0.01–0.02, correspond to around an 11–22% performance difference per decade of life, a discrepancy that we feel is likely to be unacceptable to users in the absence of other considerations.

Our findings demonstrate that CXR models do learn to encode age, despite not being trained to do so. In addition, the performance of the models varies systematically with age, exhibiting unfairness. However, it is not possible to infer from these observations alone that the encoding of age is a driver of age-related unfair performance— which would be required for shortcut learning.

### Intervening on attribute encoding affects fairness metrics

In order to test the degree to which such encoding may drive unfairness via shortcut learning, we performed an intervention that varies the amount of age encoding in the feature extractor and assessed the effect of this intervention on model fairness. We refer to this analysis as Shortcut Testing or ShorT. Multiple techniques can be used to perform this intervention (e.g., group-DRO[15], data sampling or reweighting). Based on prior work on adversarial learning[10,16–19], we selected to vary the scaling of the gradient updates from the age head in a multitask learning paradigm (see "Methods"). We focus here on results for the Effusion label (Fig. 2a); similar results were obtained for Atelectasis and Abnormal (Supplementary Fig. 2).

We were able to vary the amount of age information encoded in the feature representation across a wide range of MAE values, covering the region between the upper and lower bounds on age prediction error (Fig. 2b). By plotting the fairness of the resulting models against performance, we compared the impact of altered age encoding on these critical properties (Fig. 2c). The ideal model would have high performance and low separation, in the top left corner of the scatter plot.

In this case, we found that increasing age encoding relative to baseline (purple dots in Fig. 2c) does not noticeably affect model performance. Reducing the age encoding (green dots) appeared to slightly improve fairness properties but at the cost of reducing overall model performance. We quantitatively analyze this effect below.

### ShorT efficiently detects shortcut learning

By definition, shortcut learning is driven by correlations between an attribute such as age and condition prevalence and appearance. While not identical, the distribution of ages of patients with and

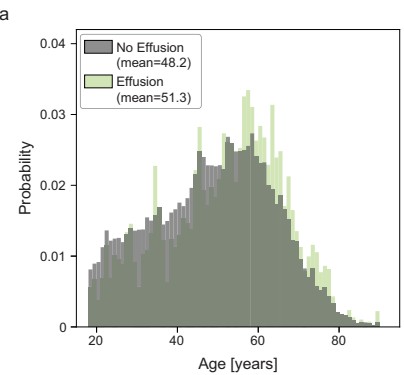
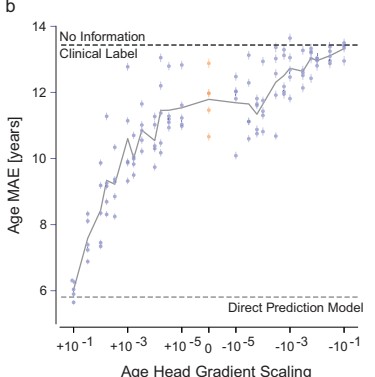
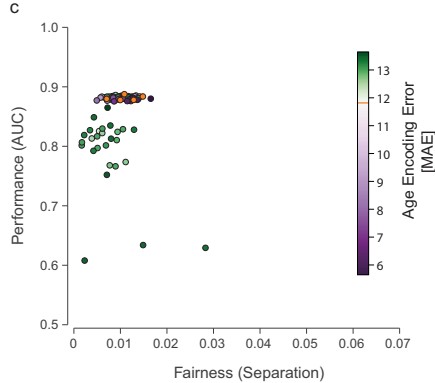

**Fig. 2 | Intervening on age encoding using multitask learning. a** The distribution of ages for positive (light green) and negative (gray) examples for Effusion in the training set. **b** The effect of altering the gradient scaling of the age prediction head on age encoding (as determined by subsequent transfer learning). For large positive values of gradient scaling (left), the models encoded age strongly, with a low MAE that approached the performance of a dedicated age prediction model (empirical LEB). For large negative values of gradient scaling (right), the age prediction performance of the multitask model approached the empirical UEB. Baseline models (with zero gradient scaling from the age head, equivalent to a single task condition prediction model) are shown in orange. Each dot represents a model trained (25 values of gradient scaling times 5 replicates), with error bars denoting 95% confidence intervals from bootstrapping examples ($n = 17,723$) within a model. **c** Fairness and performance of all replicates ($n = 125$). The degree of age encoding by the particular replicate is color-coded, with purple dots denoting more age information and green dots less information, than the baseline model without gradient scaling (in orange, overlapping with the purple dots in this case). Source data are provided as a Source Data file.

without effusion in the NIH dataset is quite similar, with a mean age of 48.2 for patients without effusion and 51.3 for patients with effusion (Fig. 2a).

We therefore applied the ShorT method to datasets in which this correlation was strengthened or weakened, simulating the results expected in contexts in which shortcut learning is respectively more or less likely. This was achieved by simple subsampling of the training data to create a biased dataset and a balanced dataset (see "Methods"). We expect that a greater correlation (biased dataset) will lead to a strong pattern of shortcut learning, while a weaker correlation (balanced dataset) should not lead to shortcut learning.

In the biased dataset, the correlation between age and the effusion label was artificially strengthened by preferentially dropping examples of older patients without effusion or younger patients with effusion. We sought to induce an approximate decade of difference between classes; owing to the inherent stochasticity of this method, the actual gap was 11.2 years (Fig. 3a). When trained on this perturbed dataset, baseline models without additional age heads represented age more strongly (9.18 years age MAE for models trained on the biased dataset, vs 11.8 years for models trained on the full dataset); however altering the gradient scaling of the age head in multitask models still resulted in a wide range of age encoding strengths (Fig. 3b). Clinical task performance was similar, albeit slightly higher in the biased dataset (mean AUC 0.901 vs 0.882 in the original dataset). This is to be expected since the separation by attribute creates further information that can be used to make more accurate predictions. However, the fairness of the models was degraded strikingly (Fig. 3c). This disparity could be obviated to some degree by gradient reversal for age—with separation approximately halved in models with poorer age representation, with only a slight decrease in overall model performance (green cluster).

On the other hand, removing age differences due to prevalence (the balanced dataset, Fig. 3d–f) resulted in models that performed at a similar level to the baseline model (mean AUC 0.883 in the balanced dataset, 0.882 original), and these were fairer than models trained on the original dataset.

We found similar results on Atelectasis and Abnormal labels when using biased datasets (Supplementary Fig. 3), with a clear pattern of dependence on model fairness relating to age encoding. Our results are also replicated for effusion with other fairness metrics (Supplementary Fig. 4).

In order to quantify the statistical dependence of unfairness on age encoding—a signature of shortcut learning—we calculated the Spearman rank correlation between these two variables (Fig. 4). For the original dataset (Fig. 4a), there was a small but statistically significant correlation between encoding and unfairness ($\rho = -0.224$, $p = 0.0156$), indicating the presence of shortcut learning. This was amplified in models trained on the biased dataset (Fig. 4b), indicating the presence of significantly stronger shortcutting when training with this biased dataset ($\rho = -0.668$, $p = 8.11e\text{-}17$). Conversely, the correlation coefficient in the balanced dataset was not significant (Fig. 4c, $\rho = 0.116$, $p = 0.218$), indicating no systematic impact of age encoding on fairness in models trained on this dataset (for details, see Supplementary Methods). Based on these results, our approach seems to efficiently detect shortcut learning.

## Shortcut learning cannot be identified by attribute encoding alone

On the other hand, we found that the amount of age information encoded in the model bears little relation to the fairness of the model when comparing datasets (Fig. 4, compared directly in Supplementary Fig. 5). For an age MAE of 8.5–9.5 years, models trained on a balanced dataset were almost perfectly fair, with an average separation coefficient of 0.0016 (range 0.0003–0.0035, $n = 24$), corresponding to an average 1.6% disparity in performance over a decade of life. In contrast, models trained on a biased dataset had a mean separation coefficient of 0.0384 (range 0.0335–0.0461, $n = 43$), corresponding to an average 47% disparity over a decade of life. Thus, in this case, it is clear that the performance of an attribute transfer model alone is insufficient to make any predictions regarding the fairness of the model. Rather, testing directly for the effect of encoding on fairness reveals the presence of shortcut learning.

## ShorT detects shortcutting by race in cardiomegaly predictions

Using the public dataset CheXpert[20], we predicted cardiomegaly from chest X-rays as a binary outcome. Following the work in Gichoya et al.[13], we investigated whether the representation of race (self-reported binary attribute, Black or White) led to shortcutting in our model. Our results showed that shortcutting was present ($\rho = 0.469$, $p < 0.001$, Fig. 5), with fairness (here estimated via

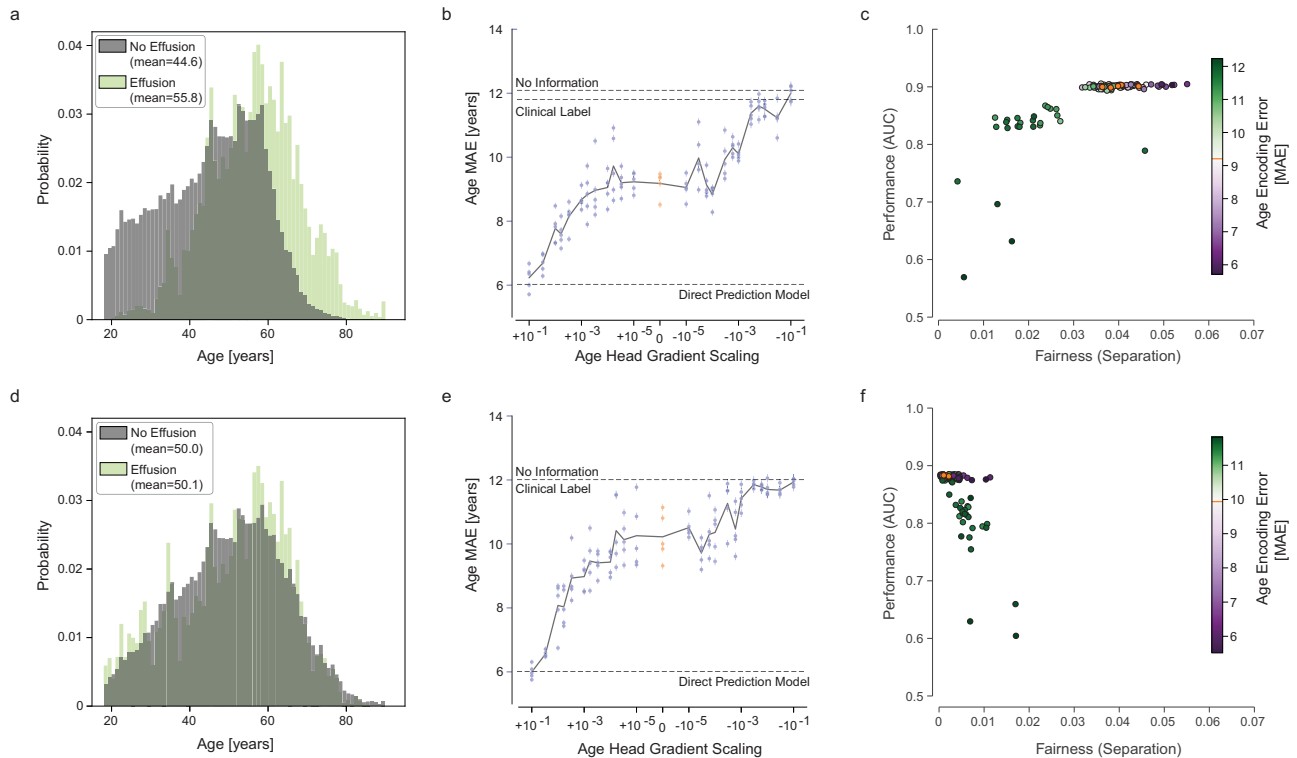

**Fig. 3 | Effect of dataset perturbation.** Results are presented in the same manner as for Fig. 2 but for a subsampled dataset inducing a larger age disparity between classes biased dataset (**a**–**c**, *n* = 15,148) and a balanced dataset (**d**–**f**, *n* = 16,093). Source data are provided as a Source Data file.

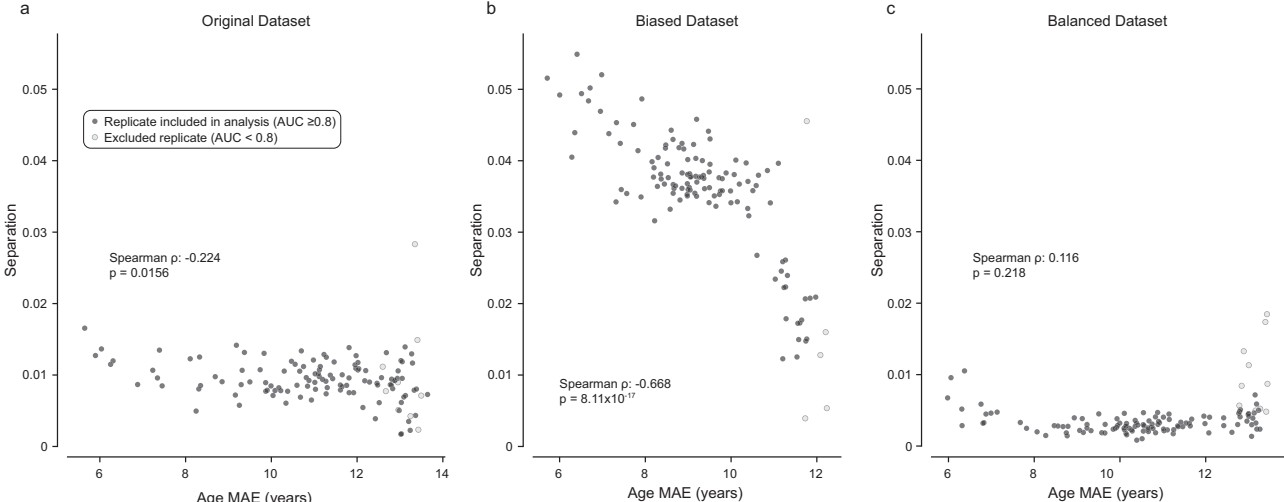

**Fig. 4 | Quantifying the systematic effect of age encoding on fairness.** In each panel, replicates are shown as individual points (*n* = 125), with age mean absolute error (MAE) on the *x*-axis and separation on the *y*-axis. Note that increasing values on the *x*-axis correspond to a reduction in age encoding; increasing values on the *y*-axis denote worsening fairness. Therefore, a correlation between encoding and unfairness would be observed as a negative correlation. **a** Original dataset. For this dataset, models which encoded age to a lesser degree (right) tended to be slightly more fair (lower separation), as reflected by a statistically significant Spearman rank correlation (two-sided). **b** The same analysis for the biased dataset. In this case, the correlation between age encoding and unfairness is markedly strengthened. **c** Balanced dataset. There was no significant correlation between age encoding and unfairness in this dataset. In all analyses, replicates with unacceptable performance (AUC < 0.8, arbitrary threshold) were excluded (open dots). Source data are provided as a Source Data file.

equalized odds) depending strongly on the model's encoding of race (estimated by the AUROC of race prediction). Contrary to the biased dataset for Effusion, we however note that there is no apparent trade-off between the model's clinical performance and fairness. In this case, ShorT provides fairer but performant alternatives to the original model in addition to the detection of shortcutting (purple dots in the top left corner of Fig. 5b).

## Beyond shortcut learning: acne prediction in a dermatology model

Lastly, we applied our approach to a multiclass prediction model in dermatology (see Supplementary Methods) similar to that published in ref. 8, examining a single class using a binarised form of our analysis. The most common condition in this dataset is Acne, which is strongly correlated with age (18.6-year difference in mean age between patients

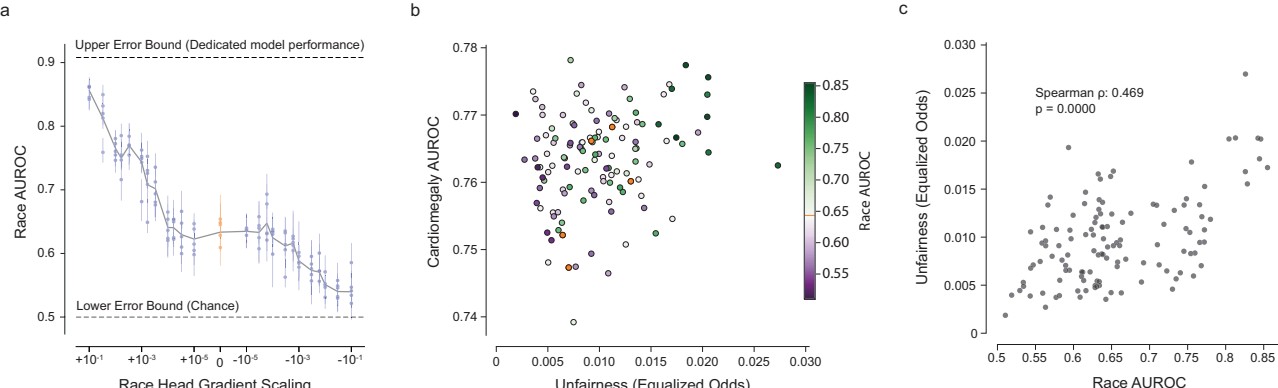

**Fig. 5 | Unfair model performance resulting from shortcut learning in a cardiomegaly classifier. a** The effect of altering the gradient scaling of the binary race prediction head on race encoding (as determined by the race prediction AUROC of subsequent transfer learning). Each dot represents a model trained (25 values of gradient scaling times 5 replicates), with error bars denoting 95% confidence intervals from bootstrapping examples within a model ($n = 3818$ independent patients). **b** AUC vs fairness (equalized odds) plot for Cardiomegaly. There exist models that are fairer and as performant as the baseline model (orange dots). **c** ShorT analysis demonstrates that unfairness is significantly correlated with race encoding in this example (two-sided Spearman correlation, $n = 123$ technical replicates). Source data are provided as a Source Data file.

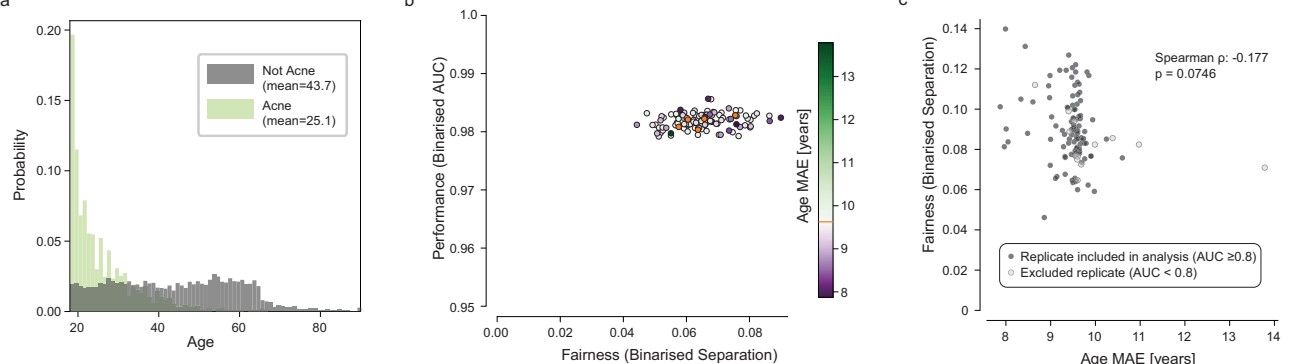

**Fig. 6 | Unfair model performance not resulting from shortcut learning in a dermatology classifier—despite strong attribute-condition correlation. a** Age distribution for examples with Acne (light green) and all other conditions (gray) in the training set. Note the significantly lower mean age for patients with Acne, as would be expected clinically. **b** AUC vs separation plot for Acne. AUC is binarised by using the score for Acne vs all; separation is calculated on a binarised prediction (top1). **c** ShorT analysis does not demonstrate that separation is significantly correlated with age encoding in this example (two-sided Spearman correlation). Source data are provided as a Source Data file.

with Acne vs other conditions; Fig. 6a). The multiclass model represents age strongly, with a mean age MAE of 9.58 years across five baseline replicates, compared to an experimental LEB of 7.32 and experimental UEB of 13.29 years (Supplementary Fig. 7).

Baseline models with no gradient updates from the age head show clear discrepancies in predicting the presence or absence of Acne (separation range 0.0576–0.0755, Fig. 6b, orange dots). This corresponds to a separate differential of up to 53% per decade.

However, despite the bias in the training set, strong age encoding in the model, and unfair performance, we found that varying the amount of age encoding did not result in a systematic change in fairness properties in this case (Fig. 6c, $\rho = -0.177$, $p = 0.0746$). Although the models are considerably less sensitive for Acne in older patients, the specific cause for this does not appear to be shortcut learning. There are a variety of other mechanisms which can lead to unfair performance, discussed below. However, multitask learning may still prove valuable in identifying models with high performance and better fairness properties.

## Discussion

Shortcut learning poses significant challenges for machine learning in healthcare, where predictions based on spurious correlations pose significant concerns regarding safety[21,22] and fairness[3,4,12]. However, identifying whether shortcut learning is responsible for model unfairness is challenging, especially when sensitive attributes such as age may be causally linked to the clinical task.

In this paper, we propose a practical method to directly test for the presence of shortcut learning during routine fairness assessment of clinical AI systems when attributes might be causally related to the outcome. We show that the degree to which models encode for a sensitive attribute—which has previously been suggested as an analytical approach[13,23]—is a poor measure of the degree to which that attribute may be used. This supports previous work demonstrating that the presence of sensitive attribute encoding by a model may arise incidentally[14]. Rather, we focus not on the encoding itself but on the degree to which the fairness of a model's predictions depend on such encoding.

While our approach is primarily designed to investigate shortcut learning, a useful by-product is that it creates a family of models mitigated with varying degrees of gradient reversal. However, the particular choice of mitigation strategy will depend on the dataset and task, and more complex strategies may prove more effective or be more feasible[24]. We also demonstrate that our method can indicate when shortcut learning is unlikely responsible for model unfairness,

which should prompt the exploration of alternative mitigation strategies.

Prior to applying our method, it is critical to select an appropriate fairness criterion. Fairness criteria are commonly classified into independence, separation and sufficiency[7]; the choice of metric will depend upon the particular clinical task, as well as the wider societal context[25]. In this study, we selected separation since age may be causally linked to many of the CXR findings our models were trained to predict, making independence an inappropriate choice. Selection of an appropriate metric also requires a deep understanding of how bias and inequity may be present in clinical environments and datasets. We therefore recommend consultation with subject experts, patient groups, and literature review; to identify plausible links between sensitive attributes and the clinical prediction target. This is likely to be improved by participatory problem formulation[26] but remains a challenging and open problem in the field to address comprehensively. We also note that multiple fairness metrics can be estimated and their correlation with the encoding of the sensitive attribute be compared. This might provide further information on which mitigation might help or select a multi-dimensional trade-off[27]. Although demonstrated in the context of separation, our framework is equally applicable to other fairness metrics.

Next, we would advocate for an initial exploratory analysis of the data, focusing on correlations that may be present between sensitive attributes and the prediction class. Such correlations are likely to drive shortcut learning. Intuitively, the greater the correlation between sensitive attribute and prediction target, the more likely it is that shortcut learning will occur. However, this is likely to be true only for similar datasets within a single task. For the prediction of Effusion on CXR, there is a clear pattern of increasing shortcut learning as the age-effusion correlation increases in subsampled datasets (Fig. 4, Supplementary Fig. 5). However, shortcut learning is still observed in the original dataset, in which there is only a small difference in age between the positive and negative classes. In the case of Acne prediction in a dermatological model, no shortcut learning was observed despite a far greater correlation between age and the condition. Therefore, correlation analysis alone is insufficient to detect shortcut learning.

Another approach that has been advocated is to investigate the degree to which models encode the sensitive attribute. As has been shown for multiple sensitive attributes[13,14], we demonstrated that clinical AI models were able to encode information about age or race, despite not being explicitly trained to do so. Therefore, providing the model with information about the age or race of a patient (as an auxiliary input) is not required for shortcut learning to occur.

Such encoding is often assumed to demonstrate that the model has learned to represent attributes so as to use them as shortcuts for predictions. However, while the presence of encoding is necessary for shortcut learning to occur, it does not provide conclusive evidence that models are basing diagnostic decisions on the encoded information using shortcut learning[14]. Our results demonstrate that the degree to which models encode sensitive information was not predictive of the fairness of the models in either the CXR or dermatology tasks. To our knowledge, this has not previously been empirically demonstrated in medical AI.

We therefore developed a method to directly test whether unfair performance is driven by the observed encoding of the sensitive attribute. We demonstrate this by varying the strength of this encoding using an additional demographic prediction head with variable gradient scaling. Models whose fairness did rely on the sensitive attribute will be systematically affected by changes in the degree of sensitive attribute encoding; models in which the encoding is incidental should not be affected in the same manner. Our approach does not quantify the strength or impact of such shortcut learning but merely whether it is statistically present in the training setup. Another important note is

that ShorT relies on an intervention that varies the encoding of the sensitive attribute in the feature extractor (here, gradient scaling). If the intervention does not cover a wide range of age encoding levels (between lower and upper error bounds), the correlation would not be reliable. On simulation data (see Supplementary Note 1), we observed that there could be a high variance in the model's encoding of the sensitive attribute based on random factors such as the random seed[28] when the sensitive attribute and the outcome had a similar signal-to-noise ratio. Therefore, verifying that the intervention consistently modifies the encoding of the sensitive attribute is needed before estimating its relationship with model fairness. Similarly, ShorT depends on the evaluation of a selected fairness metric. If this evaluation is underpowered (e.g., some subgroups being small) or highly variable, ShorT might provide misleading results. This concern is broadly applicable to any fairness evaluation. Future work could consider how to incorporate variance or confidence intervals of fairness evaluations in the formulation of ShorT.

Rather than imposing a novel or particular architecture, our approach involves the addition of a demographic prediction head to the model under investigation in order to generate a family of similar models with altered reliance upon attribute encoding. This family of models is then used primarily to define whether shortcut learning is occurring; the multihead models are not necessarily intended to be used instead of the base model.

We note that even if shortcut learning is not detected, encoding may present intrinsic ethical concerns and potential for misuse. Since we do not add the demographic head to the production model, our method reduces the potential for misuse of this information at deployment. Where one of the alternative models is found to have substantially better fairness properties, it could be substituted for the base model; we would advocate for the removal of the demographic prediction head after training in this circumstance so as to avoid any potential for misuse.

Our proposed mitigation approaches (subsampling[29,30], gradient reversal) eliminate correlations between sensitive attributes and outcomes or mitigate their effect on model training. This may seem counterintuitive, particularly where sensitive attributes are thought to be causal drivers of disease. Nevertheless, our framework allows practitioners to identify when such mitigation is desirable by analyzing the consequences of the trade-off between model performance and fairness. We however note that the selection of a specific model should be informed by domain knowledge and multiple other considerations (non-exhaustive list) such as utility, usage, potential distribution shifts[31] and downstream societal factors.

Shortcut learning is usually assumed to involve the encoding of, and reliance upon, spurious or non-causal correlations[10]. As we show, this need not be the case. Shortcut learning may also occur in cases where a sensitive attribute is strongly linked to the clinical task via known, biologically plausible mechanisms. Models may use attribute-clinical correlations appropriately, but these correlations do not generalize to deployment environments. It is also possible that models over-weight the importance of the attribute at the expense of clinical evidence that is more directly predictive, resulting in stereotyping. We chose to focus our study on age for two reasons. First, age is known to be strongly linked to disease risk across a variety of conditions. Second, age is grounded as an objective attribute rather than being socially constructed[29]. These two considerations suggest that age information may be useful in a disease prediction task, making disentangling shortcut learning from the appropriate use of input features a more difficult task. For attributes not known to be linked to disease risk or where attributes are considered to be social constructs, any use of the attribute in the prediction task may be likely to represent a shortcut. In such cases, it may be more appropriate to select demographic parity (independence) as a fairness criterion.

Where shortcutting does occur, there are multiple approaches that can be used to mitigate this particular effect. In our work, we find that gradient reversal can ameliorate, but not fully obviate, the effect of a biased dataset. Balancing the dataset may be an effective strategy, where feasible; we find that balancing the data leads to fair model performance at no cost to overall performance for CXR (see Supplementary Fig. 5). Other approaches[24,32,33] could also be considered, although recent work suggests that many strategies improve fairness only by degrading the performance of the model[34]. Regardless of the mechanism, ensuring models are as fair as possible remains a vitally important and unsolved challenge for machine learning.

When we applied our framework to a dermatology application, we did not identify a clear pattern of shortcut learning, despite unequal model performance by age. This demonstrates a case in which model performance is not fair; the label is strongly correlated with the sensitive attribute, and the sensitive attribute is encoded by the model. However, the encoding does not appear to be the (main) source of the unfair performance. In this case, unfairness might be caused by other factors, such as:

- Different presentations of the same condition. For instance, the typical pattern of hair loss for females with androgenetic alopecia differs from that of males[35]. Where presentations differ, unfairness can occur due to inadequate sample size for specific subgroups or if the appearance of the condition is more "difficult" to identify for some groups. Potential solutions include obtaining more examples of these presentations, upweighting losses for difficult examples, or approaches such as focal loss[36]. There is also evidence that longer training times encourage the learning of more difficult examples[37,38].
- Differences in the quality of the label or the use of proxy labels to approximate underlying disease. This is extensively discussed in refs. 3,39, with potential mitigations.
- Differences in the quality or missingness patterns of the data. Multiple causes of unfairness in this regard are described in ref. 1.

Lastly, shortcut learning, where present, does not guarantee the absence of other sources of unfair performance.

In our study, we focused on disease presence as a binary outcome. It is possible that disease severity or subtype distribution may also be correlated with sensitive attributes, resulting in other forms of shortcut learning. Further work is required to extend our framework to account for these considerations. Similarly, our work considered age as a single attribute of interest. In principle, this method may be readily applicable to an intersectional analysis[40], although practically, there

may be challenges around model convergence. In addition, ShorT relies on the availability of demographic data, and future research should be performed in cases where demographic data is unobserved[41].

Finally, algorithmic fairness is a set of mathematical formulations, and model behavior should be considered in the broader context of health equity, the entire clinical system and its interaction with society, rather than just focusing on model behavior given a defined dataset[42]. In this broader context, it would be informative to compare human to AI performance and fairness; and to consider modeling the therapeutic or clinical consequences of unfairness in diagnostic predictions (for example, with decision curve analysis). We however believe that the identification and mitigation of shortcut learning, as demonstrated by our approach, paves the way for more fair medical AI.

## Methods
All images and metadata were de-identified according to Health Insurance Portability and Accountability Act (HIPAA) Safe Harbor before transfer to study investigators. The protocol was reviewed by Advarra IRB (Columbia, MD), which determined that this retrospective study on de-identified data did not meet the definition of human subjects research and was hence exempt from further review.

To identify shortcut learning and how it relates to the encoding of sensitive attributes by the model, we define multiple quantities: (1) the encoding of the sensitive attribute, (2) fairness metrics, and (3) shortcut testing (ShorT), i.e., the correlation between the encoding of the sensitive attribute and fairness metrics. We demonstrate our proposed approach using binary prediction tasks for findings in an open-source chest X-ray (CXR) dataset. The approach is then applied to a multiclass diagnosis task in dermatology.

### Datasets, tasks and models
For Chest X-Ray (CXR) experiments, models were trained using the NIH Chest X Ray dataset[43] to predict a single binary condition label (Fig. 7a Condition Prediction) using a cross entropy loss. We investigated the effusion, atelectasis and abnormal findings. Model performance was estimated by computing the Area Under the Receiver Operator Curve (AUROC). The feature extractor was a ResNet $101 \times 3$ architecture initialized from BiT checkpoints[44]. We used an Adam optimizer with a constant learning rate, and optimal hyperparameters were determined for each class of model before training (see Supplementary Methods). Each model was trained from five different random seeds, with results presented in terms of average and standard deviation across seeds. The code was written in Python v3.9, using Tensorflow[45] v1.15, pandas v1.1.5[46], numpy 1.23.5[47], scikit-learn 1.0.2[48], matplotlib 3.3.4[49], and statsmodels 0.12.2[50].

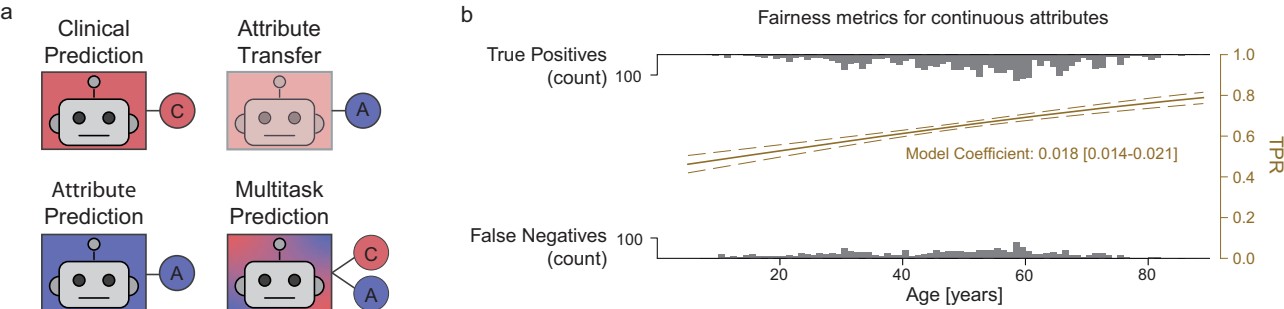

**Fig. 7 | Learning tasks and fairness metrics. a** Outline of the four learning tasks described in this paper. For each task, a feature extractor backbone (box) is used in conjunction with one or more prediction heads (circles). Clinical and Attribute Prediction tasks use a single head only; Attribute Transfer tasks use a single head with a frozen feature extractor previously trained for a clinical task. Multitask prediction models use both clinical and attribute prediction heads. **b** Logistic

Regression (LR) fit for fairness metrics. In this example, a logistic function was fitted to the predictions of the model for examples with the condition. The gray distributions represent counts of true positives (top) and false negatives (bottom) across age (x-axis). The LR model fits the probability of a true positive as a function of age (TPR, right y-axis in yellow).

## Assessing the encoding of demographic information

Similarly to previous work[13,14], we assessed the encoding of demographic information in the penultimate layer of the model by transferring the condition model to predict age. Once the condition model was trained (Fig. 7a, Clinical prediction), we froze all weights in the feature extractor and trained a linear predictor for age (Fig. 7a, Attribute Transfer) using a mean squared error regression loss. In order to aid interpretability, the performance of the transfer model was expressed as the Mean Absolute Error (MAE); this value was used as a measure of the age information content of the final layer of the feature extractor, with lower values indicating a more accurate age prediction and hence more age encoding. The results of our experiments were unchanged by using MSE as an evaluation metric. We assess the replicability of this age encoding by training 5 replicates with different random seeds.

To contextualize the obtained MAE, we trained models to directly predict age without initially training for condition prediction or freezing the feature extractor (Fig. 7a, Demographic Prediction). This provides an empirical lower error bound (LEB) for age for this dataset and model architecture. We estimated an empirical upper error bound (UEB) for age by calculating the error obtained as a result of predicting the mean age of patients in the training set for all examples in the test set (i.e., the baseline performance using the distribution of ages alone, without any image information, in this dataset).

## Assessing fairness for age as a continuous variable

To assess the fairness of a model's predictions according to age, we refer to the definitions in ref. 7 and applied group fairness metrics, quantifying independence, separation and sufficiency. Similarly to ref. 51, we expressed fairness as a function of a continuous attribute variable to avoid the need for quantizing the data[4,12]. This was achieved by fitting a univariate logistic regression (LR) model on age against the outcome of interest (Fig. 7b), implemented with Pandas and Scikit-learn[48]. Since we do not wish to assume that the clinical task is independent of age, we focus on separation—the discrepancy of error rates across subgroups, rather than independence—the discrepancy of model predictions across subgroups.

Separation was defined by fitting two LR models to the binarised model predictions, one for patients who do have the condition and one for patients who do not, equivalent to modeling the effect of age on the True Positive Rate (TPR, sensitivity) and the False Positive Rate (FPR, 1- specificity). The separation value was then calculated as the mean of the absolute values of the two logistic regression coefficients. This resulted in a metric where a score of 0 indicates that there is no monotonic relationship between age and model performance, and higher scores indicate that the TPR and/or FPR vary systematically with age. For small values of separation, our definition approximates the fractional change in performance per year of life—while numerically small, the resultant discrepancy may be large over a clinically relevant age difference. This can be calculated as $e^{s \cdot \Delta a}$, where $s$ is the separation coefficient and $\Delta a$ is the age difference. A separation value of 0.01 will thus correspond to a 10.5% change in model performance per decade; 0.02 will correspond to a 22.1% change per decade.

Our method is agnostic to the choice of fairness definition, and other equivalent formulations could be considered, such as the odds ratio of the logistic regression model, measures computing the maximum gap between subgroups[31] or metrics defined to verify the independence criteria[52]. We however note that it is best to select a metric that will not be dominated by changes in overall performance due to a possible fairness-performance trade-off (e.g., worst-case group[15]). For completeness, we report multiple fairness metrics in the Extended Data.

## ShorT: testing for shortcut learning

In prior work detecting shortcut learning, the typical assumption is that any effect of the attribute on model output is spurious[21,22].

Methods such as Group-DRO[15] or similar mitigation strategies[53] rely on this assumption and compare model performance across different groups. In the present case, we assume that a difference in performance across groups could be due to a mix of biological and shortcut-learning effects (as per our amended definition of shortcut learning). To identify shortcut learning, we hypothesize that if the model is shortcutting, intervening in its encoding of the sensitive attribute should consistently affect fairness metrics beyond the gains in performance. Our goal is hence not to perform a binary (i.e., spurious signal present compared to absent) evaluation but rather obtain a continuous modification of the encoding.

Formally, if we assume a feature representation f(X) (here the output of the feature extractor), we want to perform an intervention G such that the relationship between f(X) and the sensitive attribute A is modified between a lower bound (f(X) independent of A) and an upper bound (f(X) strongly related to A). We estimate the efficacy of our intervention with a proxy: the performance of a model predicting A from f(X). Our amended definition of shortcut learning then assesses how the relationship between A and f(X) affects the model's fairness, given a desired minimum performance level.

Based on prior work on adversarial learning[16–19], we used multitask learning to alter the degree to which age is encoded in the penultimate layer of the condition prediction model (Fig. 7a, Multitask Prediction). We trained models on both demographic information (here, age) and condition prediction tasks by using a common feature extractor with a separate head for each task. Varying the amount of age information encoded in this model was achieved by scaling the gradient updates from the age prediction head. Positive values of gradient scaling encourage the model to represent age more strongly in the final layer, whereas negative values decrease this representation by gradient reversal. For each model trained (i.e., each combination of gradient scaling and training seed), we assess population variance by bootstrapping test examples 1000 times for each model trained and report 95% confidence intervals.

We measured the effect of varying the age information encoded in the model by computing model performance, age availability (MAE of the age transfer model, as described above) and fairness metrics for each value of gradient scaling. The presence of shortcut learning is then indicated by a significant relationship between age encoding and fairness metrics. Given our choice of MAE and our formulation of separation (lower is better), we expect a negative correlation (computed via Spearman correlation coefficient) to highlight shortcut learning. For simplicity, we focus on the metrics as evaluated once across the test population for each model, as we observed that population variance was smaller than the variance across models.

## Assessing the efficacy of ShorT to detect shortcut learning

To assess how our method behaves under different bias scenarios, we alter the correlation between the sensitive attribute and the condition label in the CXR training dataset. This was achieved by randomly subsampling the training set[54], with a probability determined as a function of the patient's age and condition label (see Supplementary Methods). After resampling, we obtained two datasets: a biased dataset that introduces approximately a decade of age difference between the positive and negative classes and a balanced dataset where the distributions of ages across classes are approximately matched. Importantly, while this perturbation increased or eliminated the correlation between age and disease, there remain examples of older patients and younger patients with and without the condition, and all retained examples were not modified in any way. While this resampling strategy leads to a slightly lower number of training examples, it has the advantage of maintaining the marginal probability of diseases and avoids creating synthetic examples, which may not be realistic. We expect the biased dataset to lead to strong shortcut learning, while the balanced dataset should lead to no shortcut learning.

In Supplementary Note 1, we also generated simulated data and assessed the sensitivity of the technique when we suspected short-cutting to be absent and when we suspected it to be present.

## ShorT on other datasets

We apply ShorT on two additional datasets: chest X-rays from the public CheXpert database[20] and a proprietary dataset of dermatology images from teledermatology clinics in the United States of America[8]. Model architectures are similar to that described above. Fairness metrics and model hyperparameters are adapted to the case of a binary attribute (CheXpert) and a multiclass prediction model (dermatology). Please see the Supplementary Methods for details.

## Reporting summary

Further information on research design is available in the Nature Portfolio Reporting Summary linked to this article.

## Data availability

The NIH CXR Dataset is provided by the NIH Clinical Center and is available at https://nihcc.app.box.com/v/ChestXray-NIHCC. CheXpert is available at https://stanfordmlgroup.github.io/competitions/chexpert/. Demographic labels are available at https://stanfordaimi.azurewebsites.net/datasets/192ada7c-4d43-466e-b8bb-b81992bb80cf. The de-identified teledermatology data used in this study are not publicly available due to restrictions in the data-sharing agreement. The data is available for non-commercial purposes for an administrative fee, providing that the requesting entity can comply with applicable laws and the privacy policy of the data provider. Please contact dermatology-research@google.com who can help forward any requests to the source, with a response timeframe of maximum two weeks. The MNIST[55] data for the simulated experiments is available as a tensorflow dataset at https://www.tensorflow.org/datasets/catalog/mnist and its original version at http://yann.lecun.com/exdb/mnist/. Source data are provided with this paper.

## Code availability

The deep learning framework (TensorFlow) used in this study is available at https://www.tensorflow.org/. For medical imaging models, the training framework (Estimator) is available at https://www.tensorflow.org/guide/estimators; the deep learning architecture (bit/m-r101x3) is available at: https://tfhub.dev/google/bit/m-r101x3/1. The detailed code to model medical images is proprietary, but pseudo-code agnostic to the deep learning framework is in the supplementary material. The code for the simulated experiment using MNIST is available at https://github.com/google-research/google-research/tree/master/shortcut_testing. Scikit-learn is available at https://scikit-learn.org/stable/. We also used Matplotlib (https://matplotlib.org/) for plotting and Pandas for data analysis (https://pandas.pydata.org/).

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

## Acknowledgements

We thank Alexander D'Amour, Oluwasanmi Koyejo, Stephen Pfohl, Katherine Heller, Yun Liu, Ashley Carrick, Patricia MacWilliams, Abhijit Roy, Vivek Natarajan, Charles Lau, Jon Deaton, Dale Webster, William Isaac, Shakir Mohamed, Danielle Belgrave, Greg Corrado and Marian Croak for their contributions to this work. We also thank Lisa Lehmann and Ivor Horn for helpful discussions on the ethical implications of this work. Google funded this work.

## Author contributions

A.B., J.S., N.T., Y.L. and A.K. designed the study. A.B., J.S. and J.F. implemented the method and performed experiments. A.B., J.S., N.T., J.F. and A.K. contributed to the writing of the manuscript.

## Competing interests

All authors were funded by Google at the time of writing. The authors declare no competing interests.
