## [Peer Review File · Nature Communications]

REVIEWER COMMENTS

Reviewer #1 (Remarks to the Author):

Summary

The authors address the problem of shortcut learning, where machine learning models learn undesirable correlations in the training data, leading to poor performance for minority subgroups. They propose a method to diagnose shortcut learning, based on domain adversarial training. They evaluate their method on three tasks in ChestX-ray8 using age as the shortcut variable, finding that their method accurately diagnoses shortcut learning in the original dataset and two re-balanced datasets. The authors then use a private dermatology dataset to demonstrate that shortcut learning may not be responsible for all observed performance disparities.

Review

Major Issues

It is not clear to me why the authors chose age as a shortcut variable, as the link between age and disease risk is likely to be invariant across a wide range of environments (The authors also mention this on L291), and may even be considered causal for many diseases. It seems that race/ethnicity would serve as a better example of a shortcut to remove, as 1) it is inferrable from the x-rays as supported by the authors' citation 13, 2) it is also highly correlated with disease risk, 3) this correlation likely varies greatly across countries and regions, so is something we would want to remove.

The authors should provide a citation for their method to calculate separation for a continuous group variable (L393-405). How does it compare with previously published methods [e.g. 1]? I particularly disagree with the authors' statement that "a score of 0 indicates a perfectly fair model" (L397), as the relationship between age and the binary prediction could be complex and non-linear, and a 0 LR coefficient is not necessarily indicative of independence between the two.

In prior work in shortcut learning [e.g. 2, 3], the relative reliance on the shortcut is evaluated by examining the accuracy of the worst-case group (where groups are defined as the product of the attribute and the label sets). In contrast, the authors' separation metric (in the discrete case) would be similar to looking at the difference in performance between the best and worst-case groups. The authors should clarify how these differ, and if possible, create similar plots for the accuracy of the "worst-case" group as well.

The authors conduct their experiments on three tasks in one main dataset (NIH) where the shortcut (age) is present (original dataset), synthetically strengthened (biased dataset), and synthetically weakened (balanced dataset). The dermatology dataset shows a compelling application of the method, but it does not validate the correctness of the method. To further demonstrate the validity and flexibility of the proposed method, the authors should add an additional dataset, perhaps from a different modality, and perhaps with a different (e.g. categorical) shortcut variable.

The authors could also consider running their method on medical imaging datasets which have been identified as containing shortcuts in prior work [4, 5], or at least cite these aforementioned papers.

The proposed method is very computationally intensive, requiring a large grid of models to be trained varying the gradient scaling. In contrast, simply balancing the data, or using a method like GroupDRO [3], only require training a single additional model. The degree of shortcut reliance can then be deduced by comparing the performance of these two models.

What advantages does the proposed method have over this scheme in order to justify the additional compute required?

The authors identify shortcut learning through a negative correlation between the age MAE and separation metric. This is intuitive, and seems to work well empirically. However, its validity is not mathematically rigorously proven. Is a significant negative Spearman coefficient a necessary and sufficient condition for shortcut learning? Are there synthetic examples where this metric fails?

Minor issues:

The authors should be clearer that their method is based off of domain adversarial neural networks, which has a long history in the machine learning literature [7, 8].

The authors should state what threshold is being used to binarize the model predictions (presumably 0.5).

The authors should discuss shortcut learning in the setting where shortcuts may be unknown [2]. Could their method be adapted to this scenario?

The authors should state how many epochs the models were trained for. This is an important hyperparameter that seems to be missing from the supplement.

The authors state in the abstract that they "propose the first method to assess and mitigate shortcut learning as part of the fairness assessment of clinical ML systems". However, there are many previous works which tackle shortcut learning in the ML literature [2, 3], and the authors' citation 12 also does so in the clinical setting.

[1] Mary, Jérémie, Clément Calauzenes, and Nouredine El Karoui. "Fairness-aware learning for continuous attributes and treatments." International Conference on Machine Learning. PMLR, 2019.

[2] Liu, Evan Z., et al. "Just train twice: Improving group robustness without training group information." International Conference on Machine Learning. PMLR, 2021.

[3] Sagawa, Shiori, et al. "Distributionally robust neural networks for group shifts: On the importance of regularization for worst-case generalization." arXiv preprint arXiv:1911.08731 (2019)

[4] DeGrave, Alex J., Joseph D. Janizek, and Su-In Lee. "AI for radiographic COVID-19 detection selects shortcuts over signal." Nature Machine Intelligence 3.7 (2021): 610-619.

[5] Zech, John R., et al. "Variable generalization performance of a deep learning model to detect pneumonia in chest radiographs: a cross-sectional study." PLoS medicine 15.11 (2018): e1002683.

[6] Ganin, Yaroslav, et al. "Domain-adversarial training of neural networks." The journal of machine learning research 17.1 (2016): 2096-2030.

[7] Wadsworth, Christina, Francesca Vera, and Chris Piech. "Achieving fairness through adversarial learning: an application to recidivism prediction." arXiv preprint arXiv:1807.00199 (2018).

[8] Geirhos, Robert, et al. "Shortcut learning in deep neural networks." *Nature Machine Intelligence* 2.11 (2020): 665-673.

Recommendation

The paper tackles the important problem of diagnosing and mitigating shortcut learning in clinical deep learning models. The method proposed is intuitive, and seems to work well empirically. The paper is clearly written. I believe that the paper would be a good candidate for publication after addressing the issues raised above.

Reviewer #2 (Remarks to the Author):

The paper an in-depth study of shortcut learning and its relationship to algorithmic unfairness.

The paper is well written, and its methodology is sound. The authors motivate the definition of algorithmic unfairness, and the particular sensitive attribute. They go beyond just showing that information about this attribute is encoded by the model, and measure its correlation with the unfairness measure, separation. They then manipulate the model as a form of intervention, to assess whether this correlation can be affected.

This approach of auditing the encoding of a sensitive variable and its correlation with separation provides a clear and viable test of bias for clinical models. However, I have two primary concerns with the paper.

The first issue with the paper is the definition of the method/approach. The paper states that this is "a novel approach that represents the first practically applicable framework for studying and mitigating shortcut learning in clinical ML models". It appears to consist of several components: (1). Examining the encoding of a sensitive attribute by adding a head to a frozen predictor that predicts it; (2). Measuring the fairness of the predictor, defined here as separation; (3). Using gradient information on a 2-headed predictor to manipulate the sensitive attribute information, and examine its effect on fairness and accuracy; (4). Creating alternative datasets by varying the balance of the sensitive variable in the target classes.

It is not clear if ShorT consists of all four components. From the results in the paper it appears to. The main experiment in the paper on the CXR dataset shows that age is encoded, and the model performance varies with age, showing a form of unfairness. Yet the gradient manipulations do not have a clear effect on this age-unfairness relationship. Modifying the age encoding through gradient reversal does not appear to have a strong effect on fairness: in Figure 2c the consequence of reducing the age encoding is a decrease in model performance, while the separation metric is not affected much. The only clear relationship between the sensitive variable is obtained when both the dataset is subsampled to alter the age disparity between classes, and the gradient of age is manipulated.

This brings up the second main issue in the paper, the soundness of the mitigation strategy. Balancing the dataset with respect to the sensitive variable can have important consequences. A number of studies have shown that a subsampling approach is not a generally effective way of learning unbiased models. Especially in clinical settings where data tends to be scarce, reducing the number of training examples may not be beneficial. Some more in-depth methods for selecting examples to remove may be

warranted; others have used approaches such as influence functions to manipulate the dataset in order to mitigate bias (e.g., Removing biased data to improve fairness and accuracy, Verma et al, 2021).

The second part of the mitigation strategy, using gradient reversal in a multi-head model to reduce the encoding of the sensitive variable, could benefit from further study. In general it is not clear that the conclusions reached with a multi-head model apply to a more standard, clinically relevant model that will be trained with a single head, to predict the relevant variable.

Overall the work is strong, yet the claim of a practically applicable framework requires further clarity and justification.

ShorT: Response to Reviewers

We are grateful for the reviewers' comments and suggestions. We believe these have significantly improved our manuscript. We respond point by point to each comment (in gray) and present the changes performed to the text (in blue). A short summary of the major changes performed include:

- We performed the analysis on another chest x-ray dataset using race as a sensitive attribute. ShorT detects a significant risk of shortcutting and provides alternative models that are more fair (in Results).
- We implemented more fairness metrics (Extended Data).
- We clarified our hypotheses and definitions (Introduction, Methods, Discussion).
- We defined a simulation to illustrate the implementation of ShorT. It is open-sourced at https://github.com/google-research/google-research/tree/master/shortcut_testing

Reviewer 1

Summary

The authors address the problem of shortcut learning, where machine learning models learn undesirable correlations in the training data, leading to poor performance for minority subgroups. They propose a method to diagnose shortcut learning, based on domain adversarial training. They evaluate their method on three tasks in ChestX-ray8 using age as the shortcut variable, finding that their method accurately diagnoses shortcut learning in the original dataset and two re-balanced datasets. The authors then use a private dermatology dataset to demonstrate that shortcut learning may not be responsible for all observed performance disparities.

Review

Major Issues

1. **It is not clear to me why the authors chose age as a shortcut variable**, as the link between age and disease risk is likely to be invariant across a wide range of environments (The authors also mention this on L291), and may even be considered causal for many diseases. It seems that race/ethnicity would serve as a better example of a shortcut to remove, as 1) it is inferrable from the x-rays as supported by the authors' citation 13, 2) it is also highly correlated with disease risk, 3) this correlation likely varies greatly across countries and regions, so is something we would want to remove.

We thank the reviewer for their thoughtful and insightful comments on our manuscript. Based on this question, as well as points 3, 5 and 6, we thought it important to clarify a specificity of our work that sets it apart from typical robustness work.

Typical robustness works consider a “spurious correlation” between a variable and the outcome of interest. In the context of fairness, this would translate to a correlation between the sensitive attribute A and the label Y . If the model relies on this correlation to make its predictions, it is considered that it “shortcuts”. The terminology of a ‘spurious correlation’ implies that any observed predictive power is non-causal and should be avoided.

However, the reality in medicine can be more subtle: we may expect a certain level of relationship between a sensitive attribute and the outcome that may be driven by underlying biology. As mentioned in question 1, this would be the case for the age attribute. We sought not to remove these relationships, but to prevent the model from “over-relying” or amplifying these signals. Our work hence proposes a more nuanced definition of shortcut learning, which ShorT is able to detect. We also show that some models are unfair, but this can relate to other sources of unfairness (e.g. dermatology application) than shortcutting.

We have amended the introduction to make this clearer:

“Previous work on shortcut learning(Geirhos et al. 2020) has focussed on sensitive attributes that are likely to encode spurious correlations. In this context, any reliance by the model on the sensitive attribute may be considered to be a shortcut. However, this approach does not generalize to cases where a sensitive attribute may be, at least partly, causally related to the outcome. In this work, we posit that the effect of sensitive attributes on the model is the sum of “biological”, potentially causal effects which might improve model performance, and shortcut learning, which might be harmful. In this context, we redefine shortcut learning to be an effect of the sensitive attribute that does not considerably improve performance (as defined by the user) but affects fairness. By intervening on the degree to which a model can encode a sensitive attribute, we demonstrate the first method that assesses whether such encoding indicates the presence of shortcut learning, appropriate use of sensitive attributes, or is artefactual.”

We have also slightly modified Figure 1(b):

[REDACTED]

Figure 1: Is a model unfair due to shortcutting? (a) Examples of correct and incorrect predictions that may be influenced by age shortcut learning, in a chest x-ray application detecting Effusion, and in a dermatology predicting 27 skin condition categories, with output binarized on Acne. In these examples, the predictions are incorrect for patients with ‘atypical’ ages, raising the possibility of shortcut learning. (b) Simplified diagram illustrating how shortcutting may occur. In this schematic, the presence or absence of a particular condition y will produce changes in the image x ; we therefore wish to train a model that can predict y , given x (blue arrow). However, an attribute a , such as age, may alter both the risk of developing a given condition, as well as the image. This need not be a directly direct relationship (dotted arrows). A model may learn to predict the presence of a condition by using the attribute (red arrow). When these correlations are not considerably beneficial for model performance, we consider that this path is (at least partly) a shortcut.

Given the request in point 4, we are however happy to include another analysis which refers to race. As suggested by the reviewer, we assume that any correlation between race encoding and outcome prediction is undesirable. We refer to the CheXpert dataset and predict

cardiomegaly as a binary output. For simplicity, we focus on a binarized white/black race attribute and estimate fairness using equalized odds. Our results display that ShorT detects shortcutting in this example, and that it also provides models that are more fair while performing similarly to the baseline models.

ShorT detects shortcutting by race in cardiomegaly predictions

Figure 5: Unfair model performance resulting from shortcut learning in a cardiomegaly classifier. (a) The effect of altering the gradient scaling of the binary race prediction head on race encoding (as determined by the race prediction AUROC of subsequent transfer learning). (b) AUC vs fairness (equalized odds) plot for Cardiomegaly. There exist models that are fairer and as performant as the baseline model (red dots). (c) ShorT analysis demonstrates that unfairness is significantly correlated with race encoding in this example.

Using the public dataset CheXpert (Irvin et al. 2019), we predicted cardiomegaly from chest x-rays as a binary outcome. Following the work in Gichoya et al. (Gichoya et al. 2022), we investigated whether the representation of race (self-reported binary attribute, ‘black’ or ‘white’) led to shortcutting in our model. Our results showed that shortcutting was present ($\rho=0.469$, $p<0.001$, Figure 5), with fairness (here estimated via equalized odds) depending strongly on the model’s encoding of race (estimated by the AUROC of race prediction). Contrary to the biased dataset for Effusion, we however note that there is no apparent trade-off between the model’s clinical performance and fairness. In this case, ShorT provides fairer but as performant alternatives to the original model in addition to the detection of shortcutting (purple dots in top left corner of Figure 5b).

2. **The authors should provide a citation for their method to calculate separation for a continuous group variable** (L393-405). How does it compare with previously published methods [e.g. 1]? I particularly disagree with the authors’ statement that “a score of 0 indicates a perfectly fair model” (L397), as the relationship between age and the binary prediction could be complex and non-linear, and a 0 LR coefficient is not necessarily indicative of independence between the two.

We agree with the reviewer’s comment: our proposed metric has not been validated to prove the independence required for “separation” as per its causal definition. This metric should rather be understood as identifying a monotonic dependence, similar to the work of [Estiri et al., 2021]

which used generalized additive models (GAM) to model the effect of age on performance. We have clearly stated our metric's limitations.

We have amended our text to read:

“[a score of 0 indicates] that there is no monotonic relationship between age and model performance”

And added context regarding the importance of a specific metric (see response to point 3).

While the work in [1] is of interest, unfortunately it caters only for cases where both the attribute and the label are binary, or when both are continuous. In our case, we have a binary label and continuous attribute, which would require the method to be extended. We have included a citation to this work.

- 3. In prior work in shortcut learning [e.g. 2, 3], the relative reliance on the shortcut is evaluated by examining the accuracy of the worst-case group** (where groups are defined as the product of the attribute and the label sets). In contrast, the authors' separation metric (in the discrete case) would be similar to looking at the difference in performance between the best and worst-case groups. The authors should clarify how these differ, and if possible, create similar plots for the accuracy of the “worst-case” group as well.

We thank the reviewer for this comment and for relating our work to the “worst-case” group. Assuming a discrete or binary attribute, the maximum performance discrepancy could indeed be used as a fairness metric [a]. We have added this metric to the Extended data. We found that the “worst-group” metric is not a suitable metric as it “mirrors” the overall performance of the model, and is therefore more largely impacted by the “fairness-performance” trade-off (major effect), than by the effect of the scale gradients. We have stated this in the text.

Overall, our approach is agnostic of the fairness metric used and it could be interesting to investigate trade-offs across different fairness metrics, as inspired by Zhang et al., 2020. We have amended the text to make this point clearer:

“Our method is agnostic to the choice of fairness definition, and other equivalent formulations could be considered, such as the odds ratio of the logistic regression model, measures computing the maximum gap between groups (Schrouff et al., 2022) or metrics defined to verify the independence criteria (Mary et al. 2019). We however note that it is best to select a metric that will not be dominated by changes in overall performance due to a possible fairness-performance tradeoff (e.g. worst-case group, Sagawa et al., 2019). For completeness, we report multiple fairness metrics in the Extended Data (Figure S4).”

Figure S4: ShorT analysis of original NIH and subsampled datasets for Effusion using different fairness metrics. (Top row) Independence, as computed by the coefficient of the logistic regression between the model’s predictions and age. (Middle row) Sufficiency, as computed by the positive predictive value. (Bottom row) Maximum gap in performance across age subgroups, with age bucketed in [18,30), [30, 45), [45, 65) and [65, 100]. For all plots, an AUC threshold was set at 0.8, with replicates with an AUC value less than this being excluded from the correlation analysis.

We have also added a suggestion to compare multiple fairness metrics in the discussion:

“We also note that multiple fairness metrics can be estimated and their correlation with the encoding of the sensitive attribute compared. This might provide further information on which mitigation technique might help, or select a multi-dimensional trade-off (Zhang et al. 2020). Although demonstrated in the context of separation, our framework is equally applicable to other fairness metrics.”

[a] <https://arxiv.org/pdf/2202.01034.pdf>, to appear at NeurIPS 2022.

4. The authors conduct their experiments on three tasks in one main dataset (NIH) where the shortcut (age) is present (original dataset), synthetically strengthened (biased dataset), and synthetically weakened (balanced dataset). The dermatology dataset shows a compelling application of the method, but it does not validate the correctness of the method. **To further demonstrate the validity and flexibility of the proposed method, the authors should add an additional dataset**, perhaps from a different modality, and perhaps with a different (e.g. categorical) shortcut variable.

We agree with the reviewer that the dermatology application does not further display the correctness of the method. Based on point 1, we have added another chest x-ray dataset and analyzed the effect of race encoding on model fairness.

5. The authors could also consider running their method on medical imaging datasets which have been identified as containing shortcuts in prior work [4, 5], or at least cite these aforementioned papers.

We thank the reviewer for this suggestion. As discussed above, our work is aimed at detecting shortcut learning in the presence of a correlation that might not be spurious. The examples provided refer to site generalizability and, while it would be feasible to use our method (using site as an attribute), it would be less expensive to train on a source data and test on a target data as commonly performed in the field given that any difference in performance could be considered as “spurious”.

Re. citations: we have included these works in our reference list.

6. **The proposed method is very computationally intensive**, requiring a large grid of models to be trained varying the gradient scaling. In contrast, simply balancing the data, or using a method like GroupDRO [3], only require training a single additional model. The degree of shortcut reliance can then be deduced by comparing the performance of these two models. What advantages does the proposed method have over this scheme in order to justify the additional compute required?

We thank the reviewer for their comment. We hope that our clarification helps in understanding that the specific intervention performed to affect age encoding should be equivalent to a “knob” rather than a binary assessment of change in model performance.

One could however consider other methods than gradient scaling to perform the intervention. It is however important to understand their assumption. For instance, resampling the data would assume that the bias is related to group representation.

We have added these points in the text (see response to point 7 and the first minor issue).

7. The authors identify shortcut learning through a negative correlation between the age MAE and separation metric. This is intuitive, and seems to work well empirically. **However, its validity is not mathematically rigorously proven.** Is a significant negative Spearman coefficient a necessary and sufficient condition for shortcut learning? Are there synthetic examples where this metric fails?

We thank the reviewer for this question and understand the concern. If we consider classical robustness or fairness work, shortcut learning is typically defined as a difference in model output across environments or attribute values. The result is then binary: a significant change in model output indicates that the model is not robust or unfair.

In the present case, we provide an adapted definition of shortcut learning when the effect of the attribute might be a mix of biological and spurious effects:

If we assume a feature representation $f(X)$ (here the output of the feature extractor), we want to perform an intervention G such that the relationship between $f(X)$ and the sensitive attribute A is modified between a lower bound ($f(X)$ strongly related to A) and an upper bound ($f(X)$ independent of A). We estimate the efficacy of our intervention with a proxy: the accuracy of a model predicting A from $f(X)$. Our amended definition of shortcut learning then assesses how the relationship between A and $f(X)$ affects the model's fairness, given a desired minimum performance level. We have added these details to the Online Methods:

“In prior work detecting shortcut learning, the typical assumption is that any effect of the attribute on model output is “spurious” (DeGrave et al. 2021; Zech et al. 2018). Methods such as Group-DRO (Sagawa et al. 2019) or similar mitigation strategies (Liu et al. 2021) rely on this assumption and compare model performance across different groups. In the present case, we assume that a difference in performance across groups could be due to a mix of biological and shortcut learning effects (as per our amended definition of shortcut learning). To identify shortcut learning, we hypothesize that if the model is shortcutting, intervening on its encoding of the sensitive attribute should consistently affect fairness metrics beyond the gains in performance. Our goal is hence not to perform a “binary” (i.e. spurious signal present compared to absent) evaluation, but rather obtaining a continuous modification of the encoding.

Formally, if we assume a feature representation $f(X)$ (here the output of the feature extractor), we want to perform an intervention G such that the relationship between $f(X)$ and the sensitive attribute A is modified between a lower bound ($f(X)$ independent of A) and an upper bound ($f(X)$ strongly related to A). We estimate the efficacy of our intervention with a proxy: the performance of a model predicting A from $f(X)$. Our amended definition of shortcut learning then assesses how the relationship between A and $f(X)$ affects the model's fairness, given a desired minimum performance level.

Based on prior work on adversarial learning (Ganin and Lempitsky 2014; Ganin et al. 2016; Raff and Sylvester 2018; Wadsworth et al. 2018), we used multitask learning to alter the degree to which age is encoded in the penultimate layer of the condition prediction model (Figure 6a, ‘Multitask Prediction’). “

As this is a definition rather than a proof, we have also constructed simulated data based on MNIST images. The labels are directly related to the number present in an image (Y , whether it is smaller or larger than 5). We add a spurious signal by drawing a colored square in the top right corner of the image. The color of the square (A , red or green) can then be related to each label. We have varied multiple parameters in the simulation and found that an important limitation of our approach is the efficacy of the intervention G . We describe this in the discussion:

“We therefore developed a method to directly test whether unfair performance is driven by the observed encoding of the sensitive attribute. We demonstrate this by varying the strength of this encoding using an additional demographic prediction head with variable gradient scaling. Models whose fairness did rely on the sensitive attribute will be systematically affected by changes in the degree of sensitive attribute encoding; models in which the encoding is incidental should not be affected in the same manner. Our approach does not quantify the strength or impact of such shortcut learning, but merely whether it is statistically present in the training setup. Another important note is that ShorT relies on an intervention that varies the encoding of the sensitive attribute in the feature extractor (here gradient scaling). If the intervention does not cover a wide range of age encoding levels (between lower and upper error bounds), the correlation would not be reliable. On simulation data (see Supplement), we observed that there could be a high variance in the model’s encoding of the sensitive attribute based on random factors such as the random seed (D’Amour et al. 2020) when the sensitive attribute and the outcome had similar signal-to-noise ratio. Therefore, verifying that the intervention consistently modifies the encoding of the sensitive attribute is needed before estimating its relationship with model fairness.”

In addition, we have systematically varied the correlation between A and Y in the simulated data generation and have performed ShorT for each dataset generated ($n=20$). We obtain that ShorT leads to significant correlations between the model’s encoding of the attribute and fairness for high correlations between Y and A , but not for lower correlations. As the main task Y is easy, this is expected. We further performed a sensitivity analysis and created 50 datasets based on randomly selected low values of the correlation between A and Y and assessed how many times a significant p -value for ShorT was observed ($3/50 = 0.06 \approx 0.05$, our threshold for significance). Finally, we performed a similar analysis with high values of the correlation between A and Y and assessed how many times the p -value associated with ShorT was significant ($50/50$, Bonferroni corrected for multiple comparisons: $50/50$). We have added the information on synthetic data in the Supplement.

Finally, our method is only as good as the evaluation of the fairness metric. For instance, the CheXpert validation dataset only includes 9 black patients. Estimating the fairness of the model based on such a small sample raises multiple issues (power, variability). As this issue is not directly related to ShorT but is a general concern for fairness evaluations, we have added a comment on this in the discussion (following the section above):

“Similarly, ShorT depends on the evaluation of a selected fairness metric. If this evaluation is under-powered (e.g. some subgroups being small) or highly variable, ShorT might provide misleading results. This concern is broadly applicable to any fairness evaluation. Future work could consider how to incorporate variance or confidence intervals of fairness evaluations in the formulation of ShorT.”

Minor issues

- The authors should be clearer that their method is based off of domain adversarial neural networks, which has a long history in the machine learning literature [7, 8].

Thank you for this comment. We have made this reliance on adversarial learning more explicit:

In order to test the degree to which such encoding may drive unfairness via shortcut learning, we performed an intervention that varies the amount of age encoding in the feature extractor and assessed the effect of this intervention on model fairness. We refer to this analysis as “Shortcut Testing” or “ShorT”. Multiple techniques can be used to perform this intervention (e.g. group-DRO (Sagawa et al. 2019), data sampling or reweighting, ...). Based on prior work on adversarial learning (Ganin and Lempitsky 2014; Ganin et al. 2016; Raff and Sylvester 2018; Wadsworth et al. 2018; Geirhos et al. 2020), we selected to vary the scaling of the gradient updates from the age head in a multitask learning paradigm (see Online Methods).”

- The authors should state what threshold is being used to binarize the model predictions (presumably 0.5).
- The authors should state how many epochs the models were trained for. This is an important hyperparameter that seems to be missing from the supplement.

We apologize for these oversights. We have now added the details in the Supplement (Hyperparameter tuning and model selection):

Models were trained for 17,500 epochs and the model with highest performance on the validation data was selected. The decision threshold for each model was based on the maximum F1-score observed on validation data.

- The authors should discuss shortcut learning in the setting where shortcuts may be unknown [2]. Could their method be adapted to this scenario?

We thank the reviewer for this comment. At this point, we do not foresee a straightforward extension of the method that would accommodate unobserved demographic data. We have added this point to the discussion:

“Further work is required to extend our framework to account for these considerations. Similarly, our work considered age as a single attribute of interest. In principle, this method may be readily applicable to an intersectional analysis³⁹, although practically there may be challenges around model convergence. In addition, ShorT relies on the availability of demographic data and future research should be performed in cases where demographic data is unobserved⁴⁰. “

- The authors state in the abstract that they “propose the first method to assess and mitigate shortcut learning as part of the fairness assessment of clinical ML systems”. However, there are many previous works which tackle shortcut learning in the ML literature [2, 3], and the authors’ citation 12 also does so in the clinical setting.

Thank you. We have adjusted the wording around this point to clarify that ours is

“the first method to directly test for the presence of shortcut learning when attributes might be causally related to the outcome.”

[1] Mary, J r mie, Cl ment Calauzenes, and Nouredine El Karoui. "Fairness-aware learning for continuous attributes and treatments." International Conference on Machine Learning. PMLR, 2019.

[2] Liu, Evan Z., et al. "Just train twice: Improving group robustness without training group information." International Conference on Machine Learning. PMLR, 2021.

[3] Sagawa, Shiori, et al. "Distributionally robust neural networks for group shifts: On the importance of regularization for worst-case generalization." arXiv preprint arXiv:1911.08731 (2019)

[4] DeGrave, Alex J., Joseph D. Janizek, and Su-In Lee. "AI for radiographic COVID-19 detection selects shortcuts over signal." Nature Machine Intelligence 3.7 (2021): 610-619.

[5] Zech, John R., et al. "Variable generalization performance of a deep learning model to detect pneumonia in chest radiographs: a cross-sectional study." PLoS medicine 15.11 (2018): e1002683.

[6] Ganin, Yaroslav, et al. "Domain-adversarial training of neural networks." The journal of machine learning research 17.1 (2016): 2096-2030.

[7] Wadsworth, Christina, Francesca Vera, and Chris Piech. "Achieving fairness through adversarial learning: an application to recidivism prediction." arXiv preprint arXiv:1807.00199 (2018).

[8] Geirhos, Robert, et al. "Shortcut learning in deep neural networks." Nature Machine Intelligence 2.11 (2020): 665-673.

Recommendation

The paper tackles the important problem of diagnosing and mitigating shortcut learning in clinical deep learning models. The method proposed is intuitive, and seems to work well empirically. The paper is clearly written. I believe that the paper would be a good candidate for publication after addressing the issues raised above.

Reviewer 2:

The paper an in-depth study of shortcut learning and its relationship to algorithmic unfairness.

The paper is well written, and its methodology is sound. The authors motivate the definition of algorithmic unfairness, and the particular sensitive attribute. They go beyond just showing that information about this attribute is encoded by the model, and measure its correlation with the unfairness measure, separation. They then manipulate the model as a form of intervention, to assess whether this correlation can be affected.

This approach of auditing the encoding of a sensitive variable and its correlation with separation provides a clear and viable test of bias for clinical models.

We thank the reviewer for their positive feedback on our work.

However, I have two primary concerns with the paper.

The first issue with the paper is the definition of the method/approach. The paper states that this is "a novel approach that represents the first practically applicable framework for studying and mitigating shortcut learning in clinical ML models". It appears to consist of several components: (1). Examining the encoding of a sensitive attribute by adding a head to a frozen predictor that predicts it; (2). Measuring the fairness of the predictor, defined here as separation; (3). Using gradient information on a 2-headed predictor to manipulate the sensitive attribute information, and examine its effect on fairness and accuracy; (4). Creating alternative datasets by varying the

balance of the sensitive variable in the target classes.

1. **It is not clear if ShorT consists of all four components.** From the results in the paper it appears to.

To clarify, ShorT requires a function to estimate the encoding of the sensitive attribute and a fairness metric. Step (3) is the core of the approach, which intervenes on the encoding of the sensitive attribute and computes the correlation with fairness. Point (4) is included in the work in order to illustrate how ShorT behaves under different levels of shortcut learning, induced by dataset imbalance, but is not part of the method.

We have amended the text to make this clearer:

ShorT: testing for shortcut learning

In prior work detecting shortcut learning, the typical assumption is that any effect of the attribute on model output is “spurious” (DeGrave et al. 2021; Zech et al. 2018). Methods such as Group-DRO (Sagawa et al. 2019) or similar mitigation strategies (Liu et al. 2021) rely on this assumption and compare model performance across different groups. In the present case, we assume that a difference in performance across groups could be due to a mix of biological and shortcut learning effects (as per our amended definition of shortcut learning). To identify shortcut learning, we hypothesize that if the model is shortcutting, intervening on its encoding of the sensitive attribute should consistently affect fairness metrics beyond the gains in performance. Our goal is hence not to perform a “binary” (i.e. spurious signal present compared to absent) evaluation, but rather obtaining a continuous modification of the encoding.

Formally, if we assume a feature representation $f(X)$ (here the output of the feature extractor), we want to perform an intervention G such that the relationship between $f(X)$ and the sensitive attribute A is modified between a lower bound ($f(X)$ independent of A) and an upper bound ($f(X)$ strongly related to A). We estimate the efficacy of our intervention with a proxy: the performance of a model predicting A from $f(X)$. Our amended definition of shortcut learning then assesses how the relationship between A and $f(X)$ affects the model’s fairness, given a desired minimum performance level.

Point (4)

“[We therefore] applied the ShorT method to datasets in which this correlation was strengthened or weakened, simulating the results expected in contexts in which shortcut learning is respectively more or less likely.”

2. The main experiment in the paper on the CXR dataset shows that age is encoded, and the model performance varies with age, showing a form of unfairness. Yet **the gradient manipulations do not have a clear effect on this age-unfairness relationship.** Modifying the age encoding through gradient reversal does not appear to have a strong effect on fairness: in Figure 2c the consequence of reducing the age

encoding is a decrease in model performance, while the separation metric is not affected much. The only clear relationship between the sensitive variable is obtained when both the dataset is subsampled to alter the age disparity between classes, and the gradient of age is manipulated.

The reviewer is correct that modifying age encoding does not have a “strong” effect on the effusion model’s fairness. However, this effect is significant as estimated by a Spearman correlation, indicating that the model does shortcut. Therefore, model developers should take into account this potential for bias amplification and consider whether mitigation strategies seem appropriate. We have clarified the text to display the impact of the method (discussion):

“Our approach does not quantify the strength or impact of such shortcut learning, but merely whether it is statistically present in the training setup.”

In addition, as per Reviewer 1’s request, we have added another dataset in which the effect of shortcutting is more obvious (as we consider any reliance on the race variable for predictions to be spurious).

ShorT detects shortcutting by race in cardiomegaly predictions

Figure 5: Unfair model performance resulting from shortcut learning in a cardiomegaly classifier. (a) The effect of altering the gradient scaling of the binary race prediction head on race encoding (as determined by the race prediction AUROC of subsequent transfer learning). (b) AUC vs fairness (equalized odds) plot for Cardiomegaly. There exist models that are fairer and as performant as the baseline model (red dots). (c) ShorT analysis demonstrates that unfairness is significantly correlated with race encoding in this example.

Using the public dataset CheXpert (Irvin et al. 2019), we predicted cardiomegaly from chest x-rays as a binary outcome. Following the work in Gichoya et al. (Gichoya et al. 2022), we investigated whether the representation of race (self-reported binary attribute, ‘black’ or ‘white’) led to shortcutting in our model. Our results showed that shortcutting was present ($\rho=0.469$, $p<0.001$, Figure 5), with fairness (here estimated via equalized odds) depending strongly on the model’s encoding of race (estimated by the AUROC of race prediction). Contrary to the biased dataset for Effusion, we however note that there is no apparent trade-off between the model’s clinical performance and fairness. In this case, ShorT provides

fairer but as performant alternatives to the original model in addition to the detection of shortcutting (purple dots in top left corner of Figure 5b).

- 3. This brings up the second main issue in the paper, the soundness of the mitigation strategy.** Balancing the dataset with respect to the sensitive variable can have important consequences. A number of studies have shown that a subsampling approach is not a generally effective way of learning unbiased models. Especially in clinical settings where data tends to be scarce, reducing the number of training examples may not be beneficial. Some more in-depth methods for selecting examples to remove may be warranted; others have used approaches such as influence functions to manipulate the dataset in order to mitigate bias (e.g., Removing biased data to improve fairness and accuracy, Verma et al, 2021).

We thank the reviewer for this comment. We agree that subsampling is not suitable for all cases. In the present case, we found quite accidentally that subsampling was an efficient mitigation when we created the balanced dataset. We found particularly interesting the fact that performance was maintained even if using a smaller but balanced training set. As suggested, other resampling or reweighting techniques could be used for mitigation. We have amended the text to dampen our emphasis on subsampling as a mitigation strategy (discussion):

“Where our method reveals shortcut learning to be present, we demonstrate the use of two simple mitigation strategies (gradient reversal, subsampling) that we found improved fairness, and illustrate their impact on clinical task performance. However, the particular choice of mitigation strategy will depend on the dataset and task, and more complex strategies may prove more effective or more feasible.”

- 4. The second part of the mitigation strategy, using gradient reversal in a multi-head model to reduce the encoding of the sensitive variable, could benefit from further study.** In general it is not clear that the conclusions reached with a multi-head model apply to a more standard, clinically relevant model that will be trained with a single head, to predict the relevant variable.

We thank the reviewer for raising this confusion in our manuscript. It is first important to note that our multi-headed approach is first aimed at identifying models at high risk of shortcutting, and that the final mitigation strategy might differ from this adversarial approach. That being said, a useful by-product of the method can be a model that performs satisfactorily (as defined by its users) and that is fairer than a vanilla classifier. In this case, it is easy to remove the additional head before deployment.

We have made these statements explicit in our text:

“Rather than imposing a novel or particular architecture, our approach involves the addition of a demographic prediction head to the model under investigation, in order to generate a family of similar models with altered reliance upon attribute encoding. This family of models is then used primarily to

define whether shortcut learning is occurring; the multihead models are not necessarily intended to be used instead of the base model.

We note that even if shortcut learning is not detected, encoding may present intrinsic ethical concerns and potential for misuse. Since we do not add the demographic head to the production model, our method reduces the potential for misuse of this information at deployment. Where one of the alternative models is found to have substantially better fairness properties, it could be substituted for the base model; we would advocate for the removal of the demographic prediction head after training in this circumstance, so as to avoid any potential for misuse.”

We also provide an example of the code to perform this operation in a synthetic dataset experiment.

Overall the work is strong, yet the claim of a practically applicable framework requires further clarity and justification.

REVIEWERS' COMMENTS

Reviewer #1 (Remarks to the Author):

I thank the authors for their in-depth revisions to the manuscript. My major concerns have all been addressed.

I have a few minor recommendations:

1. I recommend the authors more thoroughly justify their choice of evaluating shortcut learning through a fairness perspective (i.e. separation) over a robustness perspective (i.e. worst-group accuracy), as is typically done in prior works on spurious correlations and shortcut learning in the ML literature. The authors briefly address this in the methods section, but I believe that the paper would be clearer if this difference to prior works was explained in the introduction.

2. I recommend the authors provide some practical guidance for model selection in the clinical setting. For example, the authors describe their method as a "knob" where you can change the level of reliance on the spurious feature, but if we had to choose a setting of the knob (i.e. a model) to deploy, which one should we choose? Is this purely based on domain knowledge, or the fairness/accuracy trade-off (e.g. some measure of social utility), or the environment where the the model is deployed (e.g. if there is a shift in the age distribution), or some additional factor?

Reviewer #2 (Remarks to the Author):

I recommend publishing the manuscript. I have a few comments about the revised version and the authors' responses to my earlier review.

1. The manuscript uses the term shortcut learning in a non-standard way. In the Geirhos et al paper cited here, "Shortcut learning typically reveals itself by a strong discrepancy between intended and actual learning strategy, causing an unexpected failure." The failures discussed in that paper refer to generalization. The authors discuss this, but then shift the failure from a generalization issue to fairness. They should clarify that they are adopting a particular alternative definition, where the failure is not a failure to generalize but rather a failure with respect to fairness.

2. In my review I stated that the mitigation strategy of gradient reversal entails constructing another predictor, with multiple heads, to control the degree of encoding of the shortcut feature. The revised text addresses this: "This family of models is then used primarily to define whether shortcut learning is occurring; the multihead models are not necessarily intended to be used instead of the base model." But in the Discussion they state that: "we demonstrate the use of two simple mitigation strategies (gradient reversal, subsampling) that we found improved fairness, and illustrate their impact on clinical task performance." This, along with the My recommendation is to de-emphasize the "Preventing" aspect of the paper, and highlight the Detecting part. Auditing for fairness effects of sensitive attributes is very important and this paper contributes significantly in this area. The mitigation aspect is weaker and could be relatively downplayed.

3. The figures are hard to appreciate and could be improved. In particular in Figure 2C the red dots are not visible, presumably because they completely overlap with the purple dots. The same holds in Figure 3C. This suggests that increasing the encoding of the age variable has no appreciable effect on performance nor fairness. I don't have a good suggestion to deal with this. You could try to make it 3D, with a separate plane for

each color, viewed from an angle to show the planes. Or just acknowledge the overlap in the caption.

ShorT - Response to Reviewers

We would like to take this opportunity to thank the Reviewers for their constructive comments and time. We respond below to the points raised, and highlight the changes made to the text (in blue).

Reviewer #1 (Remarks to the Author):

I thank the authors for their in-depth revisions to the manuscript. My major concerns have all been addressed.

Thank you, we are glad you are satisfied with our response.

I have a few minor recommendations:

1. I recommend the authors more thoroughly justify their choice of evaluating shortcut learning through a fairness perspective (i.e. separation) over a robustness perspective (i.e. worst-group accuracy), as is typically done in prior works on spurious correlations and shortcut learning in the ML literature. The authors briefly address this in the methods section, but I believe that the paper would be clearer if this difference to prior works was explained in the introduction.

Thank you for this comment. We have made the following change to the introduction:

“However, the use of information about sensitive attributes can also be harmful - in particular due to the phenomenon of “shortcut learning”¹⁰. This refers to ML models relying on spurious associations in training datasets to learn prediction rules which generalize poorly, particularly to new populations or new settings. Shortcut-based decision rules are also likely to amplify errors in ‘atypical’ examples, such as male patients with breast cancer, or melanoma in dark-skinned individuals. **While shortcut learning is typically evaluated by focusing on the performance of a model in different populations or environments¹⁰, shortcuts based upon sensitive attributes have the risk to exacerbate model unfairness, and to further health disparities. Our work hence investigates how shortcuts might affect model fairness, in addition to performance.**”

2. I recommend the authors provide some practical guidance for model selection in the clinical setting. For example, the authors describe their method as a “knob” where you can change the level of reliance on the spurious feature, but if we had to choose a setting of the knob (i.e. a model) to deploy, which one should we choose? Is this purely based on domain knowledge, or the fairness/accuracy trade-off (e.g. some measure of social utility), or the environment where

the the model is deployed (e.g. if there is a shift in the age distribution), or some additional factor?

We thank the reviewer for this great question. Everything being equal, we would typically recommend a model that encodes the attribute less. However, it is unlikely that no trade-offs will need to be performed. While the method can suggest different trade-offs between fairness and performance, the selection of a specific model is domain, task and environment specific. All factors recommended by the Reviewer should be taken into account, as well as societal downstream factors and potential system constraints (e.g. budget on resource allocation). HCI approaches could be an avenue for model selection, potentially combined with a non-interventional deployment (to estimate distribution shifts).

In this manuscript, we want to ensure that our approach is not considered as a “solution” per se, but needs to be framed into the broader context of the application. We have therefore modified and added some text around model selection in our discussion, but prefer to not provide “guidelines” that could be misinformed and/or misinterpreted.

We have added:

“Our proposed mitigation approaches (subsampling^{29,30}, gradient reversal) eliminate correlations between sensitive attributes and outcomes, or mitigate their effect on model training. This may seem counterintuitive, particularly where sensitive attributes are thought to be causal drivers of disease. Nevertheless, our framework allows practitioners to identify when such mitigation is desirable by analyzing consequences on the trade-off between model performance and fairness. We however note that the selection of a specific model should be informed by domain knowledge, and multiple other considerations (non-exhaustive list) such as utility, usage, potential distribution shifts³¹ and downstream societal factors.”

Reviewer #2 (Remarks to the Author):

I recommend publishing the manuscript. I have a few comments about the revised version and the authors' responses to my earlier review.

We thank the reviewer for their positive assessment of our work.

1. The manuscript uses the term shortcut learning in a non-standard way. In the Geirhos et al paper cited here, "Shortcut learning typically reveals itself by a strong discrepancy between intended and actual learning strategy, causing an unexpected failure." The failures discussed in that paper refer to generalization. The authors discuss this, but then shift the failure from a generalization issue to fairness. They should clarify that they are adopting a particular alternative definition, where the failure is not a failure to generalize but rather a failure with respect to fairness.

Thank you for this comment. We believe this point is similar to the one raised by Reviewer 1 (point 1). We hope our addition to the introduction makes our assessment of shortcut learning with regards to fairness clear:

“However, the use of information about sensitive attributes can also be harmful - in particular due to the phenomenon of “shortcut learning”¹⁰. This refers to ML models relying on spurious associations in training datasets to learn prediction rules which generalize poorly, particularly to new populations or new settings. Shortcut-based decision rules are also likely to amplify errors in ‘atypical’ examples, such as male patients with breast cancer, or melanoma in dark-skinned individuals. While shortcut learning is typically evaluated by focusing on the performance of a model in different populations or environments¹⁰, shortcuts based upon sensitive attributes have the risk to exacerbate model unfairness, and to further health disparities. Our work hence investigates how shortcuts might affect model fairness, in addition to performance.”

2. In my review I stated that the mitigation strategy of gradient reversal entails constructing another predictor, with multiple heads, to control the degree of encoding of the shortcut feature. The revised text addresses this: "This family of models is then used primarily to define whether shortcut learning is occurring; the multihead models are not necessarily intended to be used instead of the base model." But in the Discussion they state that: "we demonstrate the use of two simple mitigation strategies (gradient reversal, subsampling) that we found improved fairness, and illustrate their impact on clinical task performance." This, along with the My recommendation is to de-emphasize the "Preventing" aspect of the paper, and highlight the Detecting part. Auditing for fairness effects of sensitive attributes is very important and this paper contributes significantly in this area. The mitigation aspect is weaker and could be relatively downplayed.

Thank you for this comment. We rephrase this paragraph, as its intent is to highlight that we do not specifically suggest gradient reversal as the most appropriate mitigation technique:

“While our approach is primarily designed to investigate shortcut learning, a useful by-product is that it creates a family of models mitigated with varying degrees of gradient reversal. Where our method reveals shortcut learning to be present, we demonstrate the use of two simple mitigation strategies (gradient reversal, subsampling) that we found improved fairness, and illustrate their impact on clinical task performance. However, the particular choice of mitigation strategy will depend on the dataset and task, and more complex strategies may prove more effective or be more feasible²⁴. We also demonstrate that our method can indicate when shortcut learning is unlikely responsible for model unfairness, which should prompt the exploration of alternative mitigation strategies.”

3. The figures are hard to appreciate and could be improved. In particular in Figure 2C the red dots are not visible, presumably because they completely overlap with the purple dots. The same holds in Figure 3C. This suggests that increasing the encoding of the age variable has no appreciable effect on performance nor fairness. I don't have a good suggestion to deal with this. You could try to make it 3D, with a separate plane for each color, viewed from an angle to show the planes. Or just acknowledge the overlap in the caption.

Thank you for this comment. We have modified the colors of those dots to make them appear more clearly. In Figure 2c, we have added this in the caption. For 3c, the top panel displays an increase in unfairness with increasing age representation (hopefully now easier to perceive), but not the bottom panel.

“(c) Fairness and performance of all replicates. The degree of age encoding by the particular replicate is color-coded, with purple dots denoting more age information, and green dots less information, than the baseline model without gradient scaling (in orange, overlapping with the purple dots in this case).”